# The Current State of Knowledge Regarding the Genetic Predisposition to Sports and Its Health Implications in the Context of the Redox Balance, Especially Antioxidant Capacity

**DOI:** 10.3390/ijms25136915

**Published:** 2024-06-24

**Authors:** Paweł Sutkowy, Martyna Modrzejewska, Marta Porzych, Alina Woźniak

**Affiliations:** 1Department of Medical Biology and Biochemistry, Ludwik Rydygier Collegium Medicum in Bydgoszcz, Nicolaus Copernicus University in Toruń, 85-092 Bydgoszcz, Poland; p.sutkowy@cm.umk.pl (P.S.); martyna.modrzejewska@cm.umk.pl (M.M.); 2Student Research Club of Medical Biology and Biochemistry, Department of Medical Biology and Biochemistry, Faculty of Medicine, Ludwik Rydygier Collegium Medicum in Bydgoszcz, Nicolaus Copernicus University in Toruń, 24 Karłowicza St., 85-092 Bydgoszcz, Poland; m.porzych@o2.pl

**Keywords:** gene, polymorphism, exercise, antioxidant, oxidant

## Abstract

The significance of physical activity in sports is self-evident. However, its importance is becoming increasingly apparent in the context of public health. The constant desire to improve health and performance suggests looking at genetic predispositions. The knowledge of genes related to physical performance can be utilized initially in the training of athletes to assign them to the appropriate sport. In the field of medicine, this knowledge may be more effectively utilized in the prevention and treatment of cardiometabolic diseases. Physical exertion engages the entire organism, and at a basic physiological level, the organism’s responses are primarily related to oxidant and antioxidant reactions due to intensified cellular respiration. Therefore, the modifications involve the body adjusting to the stresses, especially oxidative stress. The consequence of regular exercise is primarily an increase in antioxidant capacity. Among the genes considered, those that promote oxidative processes dominate, as they are associated with energy production during exercise. What is missing, however, is a look at the other side of the coin, which, in this case, is antioxidant processes and the genes associated with them. It has been demonstrated that antioxidant genes associated with increased physical performance do not always result in increased antioxidant capacity. Nevertheless, it seems that maintaining the oxidant–antioxidant balance is the most important thing in this regard.

## 1. Introduction

### 1.1. Exercise—Health Implications

Physical exercise/effort (PE) is currently an important issue in the context of public health. A sedentary lifestyle is considered a significant pathological factor of many civilization diseases, i.e., cardiometabolic [1] and neurodegenerative diseases, as well as mental disorders (e.g., diabetes, heart attack, Alzheimer’s disease, depression) [2]. PE improves mood and physical performance, delays aging, and ameliorates cognitive function. Physically, PE is associated with increased oxygen consumption (VO_2_), which strictly affects the oxidant–antioxidant (redox) balance. Intensified cellular respiration in mitochondria increases the production of reactive oxygen species (ROSs), escalating oxidation reactions. Appropriately intensive PE may lead to a state of oxidative stress, the risk of cell damage, and, thus, inflammation. The effect of exercise depends on the intensity and duration of the effort. “Small” negative post-exercise changes can contribute to positive adaptive effects, such as increased antioxidant capacity (primarily higher activity of antioxidant enzymes) [2].

The effects of exercise on an organism are being increasingly utilized in the rehabilitation of heart diseases [3]. Moreover, PE is regarded as the most significant predictor of cardiometabolic diseases. The absence of PE is the most significant risk factor for the development of such diseases. Regular exercise is the most effective means of enhancing cardiorespiratory and metabolic health. The entire organism—all organs and physiological systems—is involved in PE [4]. Therefore, all possible kinds of genes are activated in response to the effort. This covers particularly protein genes, such as those of structural, enzymatic, and signaling proteins (muscle proteins, enzymes, receptors, cytokines, and hormones). The fundamental differences result only from the type of exercise bout (ratio of aerobic versus anaerobic efforts, endurance vs. resistance) [5]. There are parameters that connect the fields of sports and healthcare. For instance, the maximal fat oxidation rate is a predictor of metabolic flexibility, body weight loss, and endurance performance. It can be evaluated and defined not only on the basis of physical and biochemical parameters but also on the basis of genetic parameters. This is because, as is well known, everything has its origin in the genes [6].

### 1.2. Sports Genomics and the Redox Balance

It appears that the pinnacle of the human body’s capabilities has been reached in many sports. Nevertheless, the pursuit of improvement in top results remains constant. The level of professionalism in sports is on the rise, accompanied by a concurrent growth in knowledge. This has always included technique and technology, while a relatively new field of study is athletic genetics, which has significant potential for development. This is an increasingly prominent area of sports theory, representing one of many avenues in the pursuit of enhanced sports performance. For example, the use of athletic genetics could facilitate the assignment of a novice athlete to a sport discipline that is most suited to their abilities at an early stage of training. This would obviate the necessity for the athlete to undergo the arduous process of searching for a suitable sport through the conventional means of sports testing, thus allowing the athlete to be trained more expeditiously and effectively.

A study that involved monozygotic and dizygotic twins revealed that DNA may be responsible for approximately 66% of the interindividual variance in terms of sports predisposition (microsatellite variations and single nucleotide polymorphisms, SNPs) [7], and over 200 genes may even be closely related to sports performance [8]. A recent review of the literature indicates that 128 genetic markers, distributed virtually across the entire genome, may be associated with professional athletes. These include 41 polymorphisms related to endurance performance, 45 polymorphisms related to power, and 42 polymorphisms related to strength [9]. 

The first papers that appeared at the end of the twentieth century focused on only one gene that potentially strongly determines physical performance. One of the first ones was the gene for angiotensin-converting enzyme (ACE) [10]. Presently, a genome-wide association study (GWAS) is a method for evaluating genetic predisposition for sports. This is based on determining several hundred thousand to 5 million loci in one DNA sample (no assumptions regarding loci that are potentially related to the trait of interest) [11]. Sports genomics are primarily concerned with the following genes: ACE, α-actininin-3 (ACTN3), peroxisome proliferator-activated receptors (PPARs) α/γ/δ, hypoxia-inducible factor-1α (HIF1α), and endothelial nitric oxide synthase (eNOS) [5,12,13]. ACE is an indirect prooxidant enzyme, since angiotensin II, which ACE produces, increases superoxide anion radical (O_2_˙^−^) production through the activation of nicotinamide adenine dinucleotide phosphate (NADPH) oxidase (NOX) [14]. ACTN3 is a protein filament of muscle fiber involved in contraction; thus, it is not associated with the redox balance [5]. PPARs, in turn, may indirectly promote antioxidant signaling through transcriptional or post-translational activities [15]. HIF1α is a crucial subunit for transcriptional hypoxia-inducible factor, while hypoxia is a state associated with oxidative stress [16]. Likewise, eNOS, which is a source of nitric oxide (˙NO), is a kind of reactive nitrogen species (RNS) with a potent vasodilatory effect that produces peroxynitrite (ONOO^−^) in contact with O_2_˙^−^. In turn, ONOO^−^ is an ROS with a greatly strong oxidizing and vasoconstricting effect [17]. Table 1 presents a list of loci for which there is reliable evidence of a beneficial effect on physical performance.

The existing literature indicates a growing interest in the field of sports genetics. However, genes that have a direct impact on physical performance are still poorly understood. Antioxidant genes are an especially new approach to this issue, while antioxidant capacity is an essential part of post-exercise adaptation changes. Therefore, we decided to review the knowledge about genes responsible for antioxidant defense in the context of physical performance.

## 2. Body Functions during Physical Exercise

It is important to know the general physiological variations in the human organism during PE for this article. The body adjusts its physiological functions to accommodate the increased demand for energy and oxygen during physical activity. These changes include increased body temperature, respiration rate, heart rate, and blood pressure, while at a molecular level, hydrolysis and recovery of adenosine triphosphate (ATP) are intensified [18].

### 2.1. Energy Metabolism

ATP hydrolysis and recovery are essential to maintain body functions during physical activity. The energy required for muscle contractions, enabling movement, is produced through ATP hydrolysis in muscle mitochondria. For aerobic exertion, this includes the following metabolic pathways of cellular respiration: glycolysis, pyruvate oxidation, the tricarboxylic acid cycle, and oxidative phosphorylation [18]. Anaerobic bouts of exercise involve oxygen-free mechanisms such as anaerobic glycolysis and the phosphagen system (phosphocreatine utilization) based on muscle stores. The anaerobic activity involves the very rapid production of ATP, faster than under aerobic conditions, but ATP is quickly depleted because the stores of glucose and phosphocreatine in muscles are rapidly used up. Hence, anaerobic efforts allow for the generation of high power output; however, they do not last longer than 6 s. The consequence of anaerobic glycolysis is lactate production. Very intense PE with about 100% maximal VO_2_, lasting 15–30 s, involves the rapid use of glycogen stores in skeletal muscles and maximally intensified oxidative phosphorylation. The availability of O_2_ during endurance exercise and the lower intensity of the exercise allow one to continue the effort over a much longer time. Up to approximately 2 h, apart from muscle glycogen reserves, plasma glucose, plasma free fatty acids, and muscle triglycerides are used for energy production, while in addition to muscles, many other organs and body systems are involved in this process. Endurance exercise bouts of longer than 2 h are associated with a metabolic shift from carbohydrate oxidation to lipid oxidation [19].

### 2.2. Energy Resources

Physical exertion may be referred to as a stressor because it causes a reduction in fatty acid oxidation and increases glucose metabolism to cover energy demand. In contrast, fatty acid oxidation is the primary source of energy flow when resting. An instant source of energy is produced when glycogenolysis converts glycogen stored in muscle and liver cells into glucose. When the glucose concentration is too low (consumption of glycogen stores and lack of current supply of simple sugars), ATP is produced using the β-oxidation of fatty acids [18]. However, specific alterations must be considered in relation to exercise intensity and diet. Moderate-intensity exercise (40–55% VO_2max_) results in the oxidation of both lipids and carbohydrates. Hepatic glycogen stores are then mainly used, but they can also be recovered if glucose is available from digested food/drink (small intestine). An increase in the intensity of exercise results in greater involvement of skeletal muscles. PE above 75% VO_2max_ is primarily related to the consumption of muscle glycogen, which is used up very quickly during maximal and supramaximal efforts [19]. A diet rich in carbohydrates, particularly polysaccharides, can result in a high glycogen concentration within 24 to 36 h. However, a continuous supply of monosaccharides, disaccharides, and oligosaccharides is essential for prolonged endurance exercise. Interestingly, a low-carbohydrate high-fat diet increases glycogen stores but impairs aerobic metabolism and, thus, endurance exercise performance [20].

Glycogen is a glucose polymer and the main carbohydrate used for energy production. The conversion of glycogen into glucose is catalyzed by glycogen phosphorylase, which completely decomposes the glucose polymer in conjunction with a debranching enzyme [19]. To produce energy, glycogen cooperates with specific cellular components in skeletal muscles, e.g., mitochondria, myofilaments, and the sarcoplasmic reticulum. The complex of glycogen and the sarcoplasmic reticulum has been well described in fast-twitch skeletal muscle. It is believed that phosphorylase-mediated glycogen decomposition provides the availability of glucose-1-phosphate for glycolytic ATP production at this complex. Moreover, glycolytic ATP supply may facilitate Ca^2+^ reuptake into the sarcoplasmic reticulum, ensuring muscle relaxation during the contractile cycle. This is possible thanks to the interaction of glycolytic enzymes with calcium ATPases in the reticulum [21].

Depending on exercise intensity, duration, and nutritional status, adipose-derived fatty acids can be a significant or dominant origin of energy during PE. Fatty acid absorption within the cell initiates the process, while the absorption may depend on particular muscle proteins. The first step is activating fatty acid molecules with acyl-CoA synthetase found in the endoplasmic reticulum and on the surface of the outer mitochondrial membrane. The enzyme uses coenzyme A to perform ATP-dependent thioesterification of fatty acid molecules. Fatty acid combined with coenzyme A (long-chain acyl-CoA) additionally connects with a molecule of carnitine and, in this form, enters the mitochondrion. Eventually, the fatty acid undergoes oxidation inside the mitochondrion, which involves the deletion of two carbon atoms called beta and gamma (detachment of acetyl-CoA at the carboxyl end—β-oxidation of fatty acid) [22].

## 3. The Effect of Physical Exercise on the Redox Balance

The other important part of this article’s topic is the redox balance in humans, as it is a crucial part of human homeostasis during exercise. As mentioned in the introduction, changes in an organism caused by exercise result mainly from the impact on this part of the organism’s functions.

### 3.1. Mitochondrial Respiratory Chain and ROS Production

The solubility of molecular oxygen (O_2_) is up to eight-fold higher in organic solvents than in water [23]. The oxygen concentration within a cell may locally differ. Generally, the highest O_2_ concentration occurs near cellular membranes, which have physical properties similar to those of organic solvents. Mitochondria are characterized by the lowest O_2_ concentration compared to other cellular organelles, as they are central oxygen consumers. Molecular oxygen reacts with organic compounds and oxidizes them by acquiring their electrons. The complete reduction of O_2_ to water is problematic because it requires the double acquisition of two electrons (and two protons; four electrons and four protons in total) from the molecule being oxidized, and the overwhelming majority of them have paired electrons. Therefore, oxygen reacts with many compounds in the one-electron way to form O_2_˙^−^ [24]. In aerobic cells, a complete reduction of oxygen occurs in the mitochondrial respiratory chain. However, 1–4% of the oxygen consumed by mitochondria is converted into O_2_˙^−^ [24,25]. In vivo, O_2_˙^−^ reacts predominantly with iron–sulfur centers and transition metal ions. The protonated form of O_2_˙^−^, i.e., hydroperoxyl radical (HO˙_2_), which is devoid of an electric charge, penetrates cellular membranes more easily and stays longer in their hydrophobic interior (inaccessible to O_2_˙^−^), where it can initiate a reaction of lipid peroxidation. It should be noted that lipid peroxidation is an element of normal cell metabolism. Increased exposure to ROSs enhances it but does not initiate it. The attachment of another electron (and two protons) to O_2_˙^−^ produces hydrogen peroxide (H_2_O_2_). This weak oxidant reacts mainly with the thiol group of protein cysteine residues (e.g., protein tyrosine phosphatases, G proteins, some ion channels, and some transcription factors) and transition metal ions (Fe^2+^/Cu^+^). The latter group of reactions leads to the formation of the most reactive oxygen species in biological systems: hydroxyl radical (˙OH), the product of the attachment of three electrons to molecular oxygen. In the body, ˙OH reacts non-specifically with biomolecules of all major classes (low-molecular-mass compounds, proteins, lipids, carbohydrates, nucleic acids). In contrast, O_2_˙^−^ and H_2_O_2_ mainly react with enzymes that specifically break them down [26]. The enzymes catalase (CAT) and superoxide dismutase (SOD) catalyze the dismutation reaction of H_2_O_2_ and O_2_˙^−^, respectively. Peroxidases consume H_2_O_2_ to oxidize their substrates. The activity of these enzymes determines the protection of eukaryotic cells against O_2_˙^−^ and H_2_O_2_. The protection against ˙OH is mainly based on preventing its formation [24].

### 3.2. The Components of Oxidant–Antioxidant Balance

The emergence of one ROS entails the generation of others. As a result of the decomposition of O_2_˙^−^, H_2_O_2_ is formed. That, in turn, in the presence of transition metal ions, creates the possibility of the formation of ˙OH in the Haber–Weiss reaction (H_2_O_2 +_ O_2_˙^−^ + Fe^2+^/Fe^3+^ → ˙OH + OH^−^ + O_2_) or other ROSs. Physical factors such as ionizing radiation, ultraviolet radiation, photochemical reactions, and ultrasound are sources of ROSs with marginal biological significance. Intracellular sources of ROSs are much more important; they result primarily from the one-electron oxidation of reduced forms of many compounds (such as cysteine, glutathione (GSH), glucose, flavins, quinones, and nucleotides—FMNH_2_, FADH_2_, and catecholamines) by O_2_. Another source of ROSs in cells is specific enzymatic reactions. Two enzymes generating O_2_˙^−^ are particularly important here: xanthine oxidase (peroxisomes) and NOX, which are present in the plasma membrane of phagocytic cells. O_2_˙^−^ is also generated by the smooth endoplasmic reticulum and lysosomal redox chain. The reaction of O_2_˙^−^ with ˙NO produces ONOO^−^, a strong oxidant with a bactericidal effect. However, the most important cellular source of ROSs is the mitochondrial respiratory chain composed of large protein complexes: NADH-ubiquinone oxidoreductase (complex I), ubiquinone-cytochrome c oxidoreductase (complex III), and cytochrome c oxidase (complex IV). Succinate dehydrogenase (complex II) does not pump protons but provides reduced ubiquinone [24,27,28,29]. Electrons enter the chain from NADH via complex I and from succinate via complex II. Complexes I, III, and IV pump protons across the inner mitochondrial membrane, which flow back into the matrix via the ATP synthase, driving the rotor to produce ATP [29]. 

In living cells, oxidative damage caused by ROSs generated in the environment and during aerobic metabolism may lead to DNA mutations, protein inactivation, and cell death. On the other hand, ROSs may exert beneficial effects for proper cellular development and proliferation. For instance, they take part in signaling pathways, imitate and enhance the action of growth factors, or have mitogenic effects [26]. Thus, cells should have a dynamic balance between the rate of ROS formation and the decomposition rate (Figure 1). To maintain this redox balance, aerobic cells evolved defense mechanisms, which include prevention (counteracting ROS reactions with biologically important compounds), intervention (terminating free radical chain reactions), and elimination or repair (removing the products of ROS reactions with biomolecules). These mechanisms may complement and replace each other. For instance, mammalian tissues contain glutathione peroxidases (GPXs), which catalyze the reaction between GSH and H_2_O_2_, forming the oxidized form of GSH—glutathione disulfide. This disulfide may ultimately lead to protein inactivation by oxidizing the thiol groups in proteins and forming disulfide bridges (forming mixed disulfides with proteins containing thiol groups). Fortunately, GPX enzymes cooperate with glutathione reductase (GR), which recreates the reduced form of glutathione at the expense of NADPH oxidation. NADPH is regenerated by, for example, glucose-6-phosphate dehydrogenase or isocitrate dehydrogenase [24]. 

A heme-containing enzyme, cytochrome c peroxidase, located in the intermembrane space of mitochondria, decomposes H_2_O_2_ generated by mitochondrial SOD (MnSOD) with the use of the reduced form of cytochrome c. Under certain conditions, other hemoproteins (hemoglobin, myoglobin) may also protect cells against ROSs through their pseudoperoxidase activity in cooperation with low-molecular-mass antioxidants (e.g., ascorbate) [30,31]. Unfortunately, reactive ferryl forms of hemoproteins may then be formed. Flavins seem to be safer substrates for hemoproteins because their oxidized forms can be reduced by NADPH-dependent methemoglobin reductase (i.e., flavin reductase) [32].

Iron ions do not occur in free form inside or outside cells. They remain firmly bound to specific proteins (ferritin of the intestinal mucosa, transferrin—a plasma protein transporting iron, lactoferrin—a neutrophil protein) and are maintained in oxidized form—ferric ions, which cannot initiate the Fenton reaction. Haptoglobin (Hp) strongly binds free hemoglobin, while hemopexin binds heme, thus preventing lipid peroxidation catalyzed by hemoglobin and heme [24].

Copper ions in blood plasma are bound to a specific protein—ceruloplasmin—and in small amounts by albumin, transcuprein, and amino acids [33]. Intracellular copper ions are bound by metallothionein and GSH [34]. Ceruloplasmin demonstrates low SOD and ferroxidase activities, oxidizing Fe^2+^ ions to Fe^3+^ ions with the complete reduction of oxygen. 

The primary role of metallothioneins is to bind and detoxify heavy metal ions (cadmium, mercury). Still, they are also involved in the homeostasis of metals necessary for organisms (zinc and copper). A typical mammalian metallothionein protein contains approximately 20 cysteinyl residues out of 61 amino acid residues [35]. They can, therefore, react quickly with O_2_˙^−^ and ˙OH. In addition to their primary functions, some highly concentrated proteins may also act as antioxidants. For example, albumin in blood plasma binds fatty acids (potential peroxidation substrates), copper, and heme and reacts with ROSs, thereby protecting more critical macromolecules of an organism. Damaged albumin molecules undergo proteolysis and are replaced by new ones. Similarly, GSH serves as a protective target for H_2_O_2_ in cells. GPX directs the attack of H_2_O_2_ on GSH. Thus, it protects thiol groups of enzymes and prevents H_2_O_2_ from participating in the Fenton reaction. 

The fundamental differences in the physical properties of polar and non-polar cellular environments entail the existence of water-soluble antioxidants, i.e., ascorbate, GSH, uric acid, bilirubin, glucose, and pyruvate, as well as lipid-soluble antioxidants, i.e., α, γ-tocopherol, α, β-carotene, lycopene, lutein, reduced coenzyme Q, and retinol. The reactions of low-molecular-mass antioxidants with ROSs and free organic radicals are less specific than enzymatic reactions, which is not a disadvantage. These compounds are more universal defenders of the body and can play different roles. They can react with the fraction of O_2_˙^−^ and H_2_O_2_ molecules that escaped the action of antioxidant enzymes, reducing the chance of hydroxyl radical formation [24,26].

### 3.3. Exercise-Induced Changes in the Redox Balance

PE can induce changes in the concentration/activity of oxidative stress markers resulting from the increased generation of ROSs. Increased ROS generation during exercise results from enhanced oxygen demand, mainly in skeletal muscles [2,36]. Large amounts of ROSs can also be released by phagocytes that infiltrate tissue at sites of muscle damage [37]. Exercise-induced ROS generation has been shown to have both positive and negative physiological effects [37,38]. High levels of ROSs can lead to damage to cellular structures and cell death. On the other hand, ROSs play an essential role in cell signaling, for example, during muscles’ adaptation to exercise [39]. It is common knowledge that physical training improves physical fitness. Myokines, among others, are involved in this process; these are cytokines produced and released from skeletal muscles during exercise that affect metabolic and cellular processes in various tissues and organs. Some of them can affect the oxidant–antioxidant balance [40]. It has also been demonstrated that ROS-linked pathways can participate in myokine induction [39].

Numerous studies confirm the effects of exercise on the redox balance. However, the effects of this influence vary depending on the nature of changes in individual markers of the balance. ROS overproduction depends mainly on the intensity or volume of exercise [41]. For example, cycling with intensities of 50%, 60%, and 70% of the maximal VO_2_ for 10, 20, and 30 min has enhanced oxidative stress and antioxidant activities in men who lead sedentary lifestyles. However, the changes in individual oxidative stress markers observed by the researchers varied and depended on the duration of PE. SOD activity in erythrocytes, for example, increased immediately after exercise of all intensities and durations, except for an intensity of 70% of the maximal VO_2_ for 30 min, when SOD activity decreased. In the case of GPX, the only statistically significant changes consisted of a decrease in the activity of this enzyme after 20 and 30 min at all intensity levels [42]. Other studies, in turn, showed no statistically significant changes in the serum malondialdehyde (MDA) and GPX4 levels of 20 healthy young adults (men and women) and a lack of regular exercise habits after aerobic exercise at moderate intensity on a power bike (five times per week for 4 weeks). The SOD levels, on the other hand, decreased in a statistically significant manner [43]. After a 12 week maximum fat oxidation intensity exercise, however, there was a decrease in CAT activity and MDA levels and an increase in serum SOD activity in obese patients (men and women) with nonalcoholic fatty liver disease (all junior students without physical education courses). The GPX activity and total antioxidant capacity did not change in a statistically significant manner [44]. There were no changes in MnSOD, CAT, GPX, and hemeoxygenase-1 in muscle biopsy material after a 4 week training intervention consisting of high-intensity interval training in healthy volunteers [45]. A specific training program was shown to improve elite karate athletes’ oxidant–antioxidant balance. After 3 months of karate training, the authors observed an increase in SOD and CAT activity, while the MDA concentrations decreased [46]. 

Studies confirm that suitable exercise maintains physiological levels of ROSs, which allows skeletal muscle to function properly and facilitates adaptation to exercise [46,47]. Among other things, regular PE induces adaptive changes in endogenous antioxidant mechanisms [17,37,48,49]. These changes can involve both enzymatic and non-enzymatic antioxidants. It has been shown that oxidants can affect gene expression. Changes in gene expression involving ROSs have been demonstrated at the level of transcription, mRNA stability, and signal transduction. Under the influence of exercise, changes occur in SOD activity, among other things [50]. Comparing the mRNA levels of antioxidant enzymes before and after physical training, the authors showed, for example, an increase in the mRNA levels of SOD1 in peripheral blood mononuclear cells after completing a 2 week training period (jogging for 30 min 5 days per week) in healthy subjects. The mRNA levels of SOD1 and SOD2 were also found to be higher in a statistically significant manner two weeks after the completion of the training program. Before exercise, the SOD2 mRNA levels were higher in those with exercise habits compared to those without [51]. It is considered that regular exercise induces so-called “oxidative eustress” [52]. This type of oxidative stress, in contrast with “oxidative distress”, helps maintain redox homeostasis. This is possible, among other means, through the activation of transcription factors including erythroid-related nuclear factor 2 (NRF2) [52], which is considered a key regulator of antioxidant defense [53,54]. This factor regulates the expression of GR, SOD, GPX, thioredoxin reductase, hemeoxygenase-1, peroxiredoxin, metallothionein, and thioredoxin, among others. Numerous studies confirm NRF2 activation in response to exercise [55,56].

## 4. Genes Responsible for the Production of Antioxidant Capacity in the Context of Athletic Performance

### 4.1. Polymorphisms of Antioxidant Genes and Physical Performance

There are not many publicly available papers on research about the direct impact of antioxidant genes on physical performance (Table 2). Interestingly, the investigated antioxidant gene polymorphisms are not associated with improved antioxidant capacity. This covers changes leading to a weakening of the antioxidant barrier. Several SNPs of antioxidant enzyme genes have been noted to cause a decrease in the levels or activities of antioxidant enzymes. They include, among others, a valine/alanine substitution at position 9 of the amino acid chain in the mitochondrial targeting sequence (MTS) of MnSOD (Val-9Ala: NCBI, refSNP ID: rs1799725), adenine/thymine substitution at position 21 of the promoter region of the CAT gene (21A/T: NCBI, refSNP ID: rs7943316), and proline/leucine substitution at position 198 of the amino acid chain in GPX1 (Pro-198Leu: NCBI, refSNP ID: rs1050450) [57]. 

Ben-Zaken et al. [58] proved that the Ala allele in the MnSOD gene polymorphism occurs more often than the Val allele in professional athletes (N = 195) in both endurance and resistance in comparison with healthy controls (N = 240). The frequency of this allele compared to the Val allele also increases with the athlete’s training level (higher in athletes at the Olympic level than in athletes at the national level) but is not influenced by the essential type of PE (aerobic/endurance or anaerobic/resistance). MTS directs the primary translation product of MnSOD to the mitochondrion; therefore, the replacement of valine with alanine causes a change in the spatial conformation of MTS from an α-helix to a β-sheet, impairs the transport of the enzyme, and ultimately reduces the antioxidant capacity in mitochondria. The authors suggest that increased physical performance in this way results from angiogenesis, mitochondrial biosynthesis, and muscle hypertrophy, which are intensified due to higher ROS concentrations [58]. A point mutation in the MnSOD gene with a similar redox effect was also investigated by Ahmetov et al. [59]. Specifically, this is about the Ala16Val polymorphism (rs4880 C/T), which is concerned with a substitution of cytosine for thymine, resulting in a substitution of alanine for valine at position 16 of the amino acid chain in MTS. Consequently, a reduction in the mitochondrial antioxidant capacity is observed as a result of the inefficient transport of MnSOD across the mitochondrial membrane. In contrast to the findings of Ben-Zaken et al. [58], the authors of this study demonstrated that the genotype associated with reduced MnSOD antioxidant capacity is less prevalent among athletes. However, this refers only to strength athletes (N = 524), as no differences were observed between endurance athletes (N = 180) and controls (N = 917). The mentioned antioxidant gene polymorphisms were also evaluated in the context of physical performance by Akimoto et al. [57]. They examined long-distance runners (135 individuals in total, 82 men and 53 women aged 15–58 years) with the following gene polymorphisms: Val/Val, Ala/Val, and Ala/Ala genotypes of MnSOD, AA, AT, and TT genotypes of CAT, and Pro/Pro, Pro/Leu, and Leu/Leu genotypes of GPX1. The DD, ID, and II genotypes of ACE were also evaluated. The impact of a given genotype on endurance performance was assessed. This was done using the parameters of post-exercise damage in venous blood serum: aspartate aminotransferase, alanine aminotransferase, creatine kinase, thiobarbituric acid reactive substances (TBARSs, mainly MDA), as well as using a comet assay of nucleated blood cells (leukocytes). The runners were tested with a single run over a distance of 4–21 km. The study shows that only the MnSOD polymorphisms are related to aerobic capacity. The authors found statistically significant lower serum activity of creatine kinase in runners with the Ala/Ala MnSOD genotype compared to the enzyme activities in runners with the other MnSOD genotypes. The study by Ahmetov et al. suggests an opposing relationship. In their study, elevated creatine kinase activity (women) and creatinine concentration (men and women) were positively correlated in professional athletes with the MnSOD polymorphism, which results in a reduction in mitochondrial antioxidant capacity [59]. Akimoto et al. [57] were also the first researchers to examine glutathione S-transferase (GST) in the context of physical performance. They studied cytosolic GST-µ and GST-θ, which are encoded by the glutathione S-transferase M1 and T1 genes, respectively. The family of those isoenzymes (EC 2.5.1.18) plays a key role in redox neutralization (reduction) of potentially harmful metabolic products using GSH. These mainly include a wide spectrum of xenobiotics and oxidative stress products [60]. It has been found that polymorphisms of these genes, which are based on allelic deletion (null genotype), result in a higher prevalence of neoplasm [60,61], and these are the loci that the authors examined, but they found no relation between them and endurance performance [57]. Moreover, Akimoto et al. [57] revealed that haptoglobin, a serum glycoprotein that is a type of acute-phase protein, may also be involved in endurance performance. The serum TBARS concentration was lower in runners with the 1S-1S Hp genotype than in runners with the 1F-1S and 1S-2 genotypes (no statistically significant differences compared to the other genotypes: 1F-1F, 1F-2, and 2-2). Hp is responsible for binding free hemoglobin in the bloodstream, which prevents oxidative stress resulting from the Fenton reaction [62]. Hp dysfunctions are associated with a higher incidence and clinical stage of many inflammatory and autoimmune diseases [63]. For instance, the 2-2 genotype is associated with an increased risk of developing vascular complications in patients suffering from diabetes. This results from the fact that heme iron combined with this type of Hp is more susceptible to redox reactions than other Hp complexes with heme iron, which results in higher concentrations of oxidized lipids and, thus, increased levels of dysfunctional lipoproteins [64].

### 4.2. Antioxidant Adaptation as a Result of Exercise

As mentioned in the previous section, regular PE has been widely shown to increase the ability to alleviate inflammation and oxidative stress by inducing an adaptive response in the endogenous antioxidant system. This is manifested by, among other things, the positive impact of exercise on ˙NO bioavailability in the endothelium. It has been found that the ˙NO concentration increases as a result of regular physical activity in both young and elderly individuals [65], which may be accompanied at the same time by the maintained redox balance (correspondingly higher plasma antioxidant capacity) and eventually proper cutaneous microcirculatory functions in older people [66]. Another research study also points to this; 36 older (47–74 years) professional long-distance runners had better regulation of blood circulation (endothelium function) and higher plasma antioxidant capacity than 36 controls (sex-matched, untrained people aged 46–77 years, with no cardiovascular disease or any risk factors of the disease). Blood flow in the hands and feet was examined in the participants under conditions of increased temperature (44 °C), as well as ischemia and hyperemia (3 min of brachial artery occlusion), using a laser Doppler flowmetry and biochemical parameters in venous blood (left cephalic vein). Higher plasma concentrations of ˙NO in the form of its metabolites (nitrites and nitrates), mRNA of PGC-1α (PPARγ coactivator-1 α), total antioxidant capacity, and sirtuin 1, as well as higher miR29 concentrations in mononuclear cells of the blood, were found in the athletes [67]. The increase in the eNOS activity and phosphorylation of serine at the 1177 position of its amino acid chain is most probably an adaptive and vasoprotective effect of PE. The molecular changes also include increased levels of integrins (cell membrane proteins) and H_2_O_2_ in the endothelium. Exercise-induced eNOS activation is transient and reversible and is regulated in redox reactions, including the upregulation of SOD (SOD1 and SOD3) and downregulation of NOX [68].

The same can be said of myokines, which, as previously stated, play a role in the physiological adaptations that occur during exercise. The induction and release of myokines are, in part, mediated by the muscles’ production of ROSs [39]. At the same time, some myokines exhibit considerable antioxidant potential and the capacity to regulate the redox balance. For example, under ischemia/reperfusion-induced oxidative stress in the heart, brain-derived neurotrophic factor (BDNF) has been shown to reduce the concentration of H_2_O_2_, while leukemia inhibitory factor (LIF) and fibroblast growth factor 21 (FGF-21) have been demonstrated to result in a lower concentration of O_2_˙^−^. A similar effect regarding O_2_˙^−^ has been found for interleukin-6 (IL-6) in the brain and regarding ONOO^−^ for fibroblast growth factor 2 (FGF-2) in the kidneys. Concurrently, the same or other myokines contribute to the higher activities of antioxidant enzymes (SOD, GPX) and/or oxidant enzymes (NOS, NOX) [40]. In general, myokines are signaling molecules that act in a number of ways—inside muscle cells (“in situ”), including in an autocrine manner, as well as in paracrine and endocrine manners on other tissues throughout the body. They may protect against oxidative stress while also acting as modulators of metabolism via redox reactions in both physiological and pathological conditions (e.g., PE and aging, as well as cardiometabolic diseases and cancer, respectively). For example, irisin, a recently identified myokine, appears to possess the ability to mitigate and even repair oxidative damage to muscle tissue caused by the aging process in individuals who engage in regular exercise. Myokines are also involved in the regulation of muscle tissue reconstruction. In satellite cells, the Pax genes are activated, and the sequential expression of myogenic regulatory factors occurs as follows: MyoD, Myf5, myogenin, and MRF4. The result is the proliferation, differentiation, and fusion of satellite cells into new multinucleated muscle cells, processes that are dependent on a transient increase in the concentration of ROSs [68]. Furthermore, numerous animal and in vitro studies have corroborated these findings [40,68]. For instance, cardiomyocyte culture and adult mice subjected to a pro-inflammatory agent (lipopolysaccharide) have been shown to enhance the cardiac concentration of FGF-21, which resulted in increased expression of antioxidant genes (SOD-2 and uncoupling protein 3, UCP-3) and capacity via an autocrine manner [69].

Finally, sometimes, a gene seemingly unrelated to the oxidant–antioxidant balance can affect it. An example is the MYBPC3 gene, in which SNPs are associated with either the phenotype of an elite athlete’s heart or hypertrophy cardiomyopathy, depending on the gene allele. MYBPC3 encodes myosin-binding protein C3. Phosphorylation of this structural protein increases cardiac contraction, and its isoforms are also present in skeletal muscles. The polymorphisms of the gene present in professional athletes are probably related to theophylline, quinate, and decanoylcarnitine, metabolites whose level increases with aerobic capacity and that increase antioxidant capacity (theophylline and decanoylcarnitine). In turn, hypertrophy cardiomyopathy, a condition characterized by oxidative stress, is associated with ursodeoxycholate [70].

## 5. Conclusions and Future Perspectives

### 5.1. Conclusions

The importance of physical exertion in healthcare is increasingly being recognized, with evidence indicating that it is an effective means of improving metabolic and cardiorespiratory health. In this context, exercise is accorded particular attention with regard to the prevention and treatment of metabolic syndrome and its associated diseases [3,4]. In contrast, in the field of sports, novel approaches to enhancing performance are consistently being investigated. Therefore, investigating the origin of an organism’s characteristics, specifically its genes, can facilitate the realization of health benefits that extend beyond the primary objective [6].

The literature in the field of sports genomics is mainly focused on genes that regulate muscle functions during physical exertion and genes of structural proteins in muscles, especially those responsible for contraction. Most of these genes are directly or indirectly related to oxidant processes, not antioxidant ones [5,9,12,13,14,15,16,17]. Another issue is inconsistent conclusions regarding the impact of exercise on the redox balance. On the one hand, the human body needs free radicals for effective PE, as they are signal conductors between cells/tissues. In any case, the consequence of this effort is the augmented generation of free radicals. ROSs and RNSs can also potentially threaten an organism in excessive concentrations. They may contribute to severe oxidative damage of cellular components and lead to pathological changes, starting with inflammation. However, this is not the case with exercise, as the effects of PE on an organism remain within the parameters of physiological responses. The consequences of free radicals, oxidative-stress-induced microdamage, and inflammation as a result of exercise are adaptive changes in the body, which are expressed in higher antioxidant capacity. This is a condition that an organism can cope with and ultimately benefit from [36,37,38,39]. The entire effect of exercise on the human redox equilibrium depends on the intensity, type, and duration of an exercise bout. The physical fitness of a given individual is also important, as well as whether an individual is a healthy or sick person [42,43,44,45,46]. In general, however, an organism’s reactions tend to maintain the redox balance in each condition (keeping the system’s self-regulation). This should be remembered when concluding research studies on antioxidant genes in the context of physical performance. On the one hand, it has been demonstrated that the frequency of prooxidant modification in the MnSOD gene (Ala allele) is strongly and positively correlated with the training level without an association with the type of exercise [57,58]. On the other hand, there are alleles of antioxidant genes that increase antioxidant capacity and, thereby, physical performance (e.g., 1S-1S allele of the Hp gene) [57]. Maintenance of the redox balance—the cooperation between oxidants and antioxidants in an organism to keep its proper functions—is well illustrated by the endothelium’s already-mentioned regulation of blood pressure. For appropriate endothelium function, both oxidants (˙NO, H_2_O_2_) and antioxidants (higher total plasma antioxidant capacity) are required [65,66,67,68]. A comparable redox effect is observed in the case of myokines [40,69]. 

### 5.2. Future Perspectives

It should be remembered that genes interact in many different ways. Thus, an examination of particular genes in one research experiment may provide different results in another, depending on the subjects, i.e., on the genotypes included in a study. The entire genotype shapes the phenotype. SNPs are not the only possible origin of differences. Many other genetic markers should also be considered (e.g., rare mutations), as well as epigenetic features. Future studies should also involve genes linked to other sport-related predispositions, in addition to those representing exercise physiology and anatomy. For instance, genes involved in shaping personality and mental traits should be included.

## Figures and Tables

**Figure 1 ijms-25-06915-f001:**
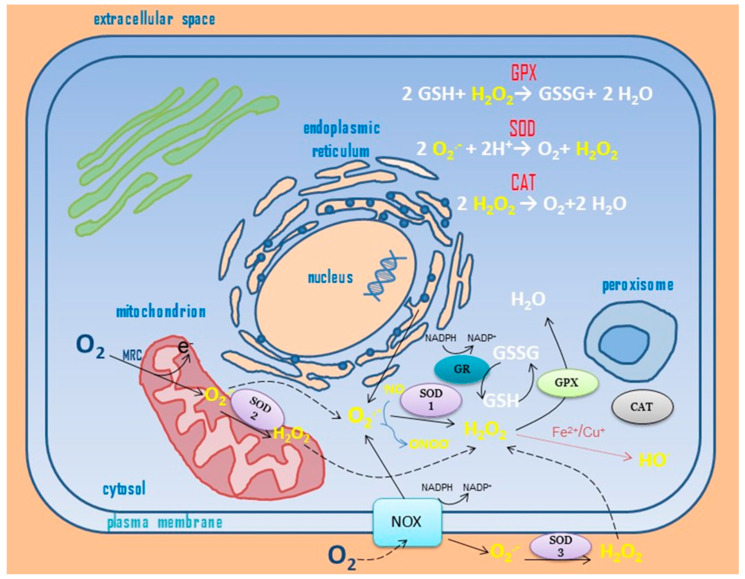
Maintaining a dynamic equilibrium between the formation and elimination of reactive oxygen species in eukaryotic cells. CAT: catalase; GPX: glutathione peroxidase; GR: glutathione reductase: GSH/GSSG: reduced/oxidized form of glutathione; H_2_O_2_: hydrogen peroxide; MRC: mitochondrial respiratory chain; NADPH/NADP^+^: reduced/oxidized form of nicotinamide adenine dinucleotide phosphate; ˙NO: nitric oxide; NOX: NADPH oxidase; O_2_: molecular oxygen; O_2_˙^−^: superoxide anion radical; ˙OH, hydroxyl radical; ONOO^−^: peroxynitrite; SOD 1–3: superoxide dismutases 1–3.

**Table 1 ijms-25-06915-t001:** Alleles that have been demonstrated to enhance physical performance [9].

Gene	Locus	Alleles	Type of Benefit
AMPD1 (Adenosine Monophosphate Deaminase 1)	1p13	rs17602729 C/T	endurance/power (C allele)
CDKN1A (Cyclin-Dependent Kinase Inhibitor 1A)	6p21.2	rs236448 A/C	endurance (A allele)/power (C allele)
HFE (Homeostatic Iron Regulator)	6p21.3	rs1799945 C/G	endurance (G allele)
MYBPC3 (Myosin Binding Protein C3)	11p11.2	rs1052373 A/G	endurance (G allele)
NFIA-AS2 (NFIA antisense RNA 2)	1p31.3	rs1572312 C/A	endurance (C allele)
PPARA (Peroxisome Proliferator Activated Receptor A)	22q13.31	rs4253778 G/C	endurance (G allele)
PPARGC1A (Peroxisome Proliferative Activated Receptor Γ coactivator 1 A)	4p15.1	rs8192678 G/A	endurance (G allele)
ACTN3 (Actinin A 3)	11q13.1	rs1815739 C/T	power/strength (C allele)
CPNE5 (Copine V)	6p21.2	rs3213537 G/A	power (G allele)
GALNTL6 (Polypeptide N-acetylgalactosaminyltransferase Like 6)	4q34.1	rs558129 T/C	power (T allele)
IGF2 (Insulin-Like Growth Factor 2)	11p15.5	rs680 A/G	power (G allele)
IGSF3 (Immunoglobulin Superfamily Member 3)	1p13.1	rs699785 G/A	power (A allele)
NOS3 (Nitric Oxide Synthase 3)	7q36	rs2070744 T/C	power (T allele)
TRHR (Thyrotropin-Releasing Hormone Receptor	8q23.1	rs7832552 C/T	power (T allele)
AR (Androgen Receptor)	Xq12	CAG repeats	strength (allele of ≥21 CAG repeats)
LRPPRC (Leucine-Rich Pentatricopeptide Repeat Cassette)	2p21	rs10186876 A/G	strength (A allele)
MMS22L (Methyl Methanesulfonate-Sensitivity Protein 22-Like)	6q16.1	rs9320823 T/C	strength (T allele)
PHACTR1 (Phosphate and Actin Regulator 1)	6p24.1	rs6905419 C/T	strength (C allele)
PPARG (Peroxisome Proliferator Activated Receptor Γ)	3p25.2	rs1801282 G/C	strength (G allele)

**Table 2 ijms-25-06915-t002:** Polymorphisms of antioxidant parameters assessed for their relationships with physical performance.

Parameter	Genotypes (Single-Nucleotide Polymorphisms)	Relationship with Physical Performance	References
Manganese superoxide dismutase (SOD 2, EC 1.15.1.1)	Val/Val, Ala/Val, Ala/Ala	Yes	[57,58,59]
Catalase (CAT, EC 1.11.1.6)	AA, AT, TT	No	[57]
Glutathione peroxidase 1 (GPX 1, EC 1.11.1.9)	Pro/Pro, Pro/Leu, Leu/Leu	No	[57]
Haptoglobin (Hp)	1F-1F, 1F-1S, 1S-1S, 1F-2, 1S-2, 2-2	Yes	[57]
Glutathione S-transferase (GST, EC 2.5.1.18)	GSTM1-, GSTM1+, GSTT1-, GSTT1+, GSTM1-T1-, GSTM1+T1-, GSTM-T+, GSTM+T+	No	[57]

Val: valine; Ala: alanine; A: adenine; T: thymine; Pro: proline; Leu: leucine.

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
