# Peer review of "The Current State of Knowledge Regarding the Genetic Predisposition to Sports and Its Health Implications in the Context of the Redox Balance, Especially Antioxidant Capacity"

_ijms, 2024, doi:10.3390/ijms25136915_

Round 1

Reviewer 1 Report

Comments and Suggestions for Authors

Dear Editor,

I was requested to review the manuscript with the title “Antioxidant Gene Polymorphisms Potentially Predisposing to Exercise” which highlights the importance of maintaining the balance between oxidant and antioxidant reactions during regular physical exercise.

Major comment: The chapters are too long and is difficult to follow sometimes, please try to divide each chapter into smaller sections or use subheadings to help readers follow the discussion more easily.

Minor comments:

1.Body functions during physical exercise: Please add more information on how glycogen stores vary with training and diet.

2.Genes responsible for the production of antioxidant capacity: Please elaborate on the significance of myokines in modulating oxidative stress during exercise.

Author Response

Dear Reviewer,

Thank you for reviewing the manuscript. We value your commitment and contribution to the paper, and your comments have undoubtedly improved its quality. Please find below your comments and our replies. All changes are highlighted in the manuscript file.

Best regards,

The authors

Comments and Suggestions for Authors

Dear Editor,

I was requested to review the manuscript with the title “Antioxidant Gene Polymorphisms Potentially Predisposing to Exercise” which highlights the importance of maintaining the balance between oxidant and antioxidant reactions during regular physical exercise.

Major comment: The chapters are too long and is difficult to follow sometimes, please try to divide each chapter into smaller sections or use subheadings to help readers follow the discussion more easily.

Authors’ Response: We agree—we have added subheadings.

Minor comments:

1.Body functions during physical exercise: Please add more information on how glycogen stores vary with training and diet.

Authors’ Response: Supplemented (the first paragraph of “Energy resources” subchapter (2.2.)).

2. Genes responsible for the production of antioxidant capacity: Please elaborate on the significance of myokines in modulating oxidative stress during exercise.

Authors’ Response: Added (the second paragraph in subchapter 4.2).

We have also updated chapter 4 with another study on SOD2 gene polymorphism and exercise implications

The other changes have been dictated by the comments of other reviewers.

Reviewer 2 Report

Comments and Suggestions for Authors

This paper is a review of the importance of oxidative and antioxidant balance in sports, exercise, and athletic performance. I find this paper to be very interesting, thorough, and excellent.

The first paragraph of the abstract and the last paragraph of the introduction describe the following

"The genes that directly influence physical performance are still poorly understood, and antioxidant capacity is an essential part of the adaptive changes that occur after exercise. Therefore, the knowledge of the genes responsible for antioxidant defense in the context of physical performance is reviewed".

However, the answers (e.g., effects of oxidative and antioxidant responses on physical and exercise performance) are not clearly stated in the abstract or in section "5. Conclusions and future perspectives". The authors should discuss and emphasize this point more carefully.

Author Response

Dear Reviewer,

Thank you for reviewing the manuscript. We value your commitment and contribution to the paper. Please find below your comments and our reply. All changes are highlighted in the manuscript file.

Best regards,

The authors

Comments and Suggestions for Authors

This paper is a review of the importance of oxidative and antioxidant balance in sports, exercise, and athletic performance. I find this paper to be very interesting, thorough, and excellent.

The first paragraph of the abstract and the last paragraph of the introduction describe the following

"The genes that directly influence physical performance are still poorly understood, and antioxidant capacity is an essential part of the adaptive changes that occur after exercise. Therefore, the knowledge of the genes responsible for antioxidant defense in the context of physical performance is reviewed".

However, the answers (e.g., effects of oxidative and antioxidant responses on physical and exercise performance) are not clearly stated in the abstract or in section "5. Conclusions and future perspectives". The authors should discuss and emphasize this point more carefully.

Authors’ Response

We have corrected the paper according to the suggestions.

We have also updated chapter 4 with another study on SOD2 gene polymorphism and exercise implications.

The other changes are dictated by the comments of other reviewers and, in our opinion, correspond with these comments.

Reviewer 3 Report

Comments and Suggestions for Authors

Review for the manuscript “Antioxidant Gene Polymorphisms Potentially Predisposing to 2 Exercise ”

In this article, the authors said that there are genetic predisposition to sports, such as the genes that promote oxidative processes, or genes associated with antioxidant processes.  

Comments

Presentation of the article is a litle beat hard to follow. Maybe more figures are useful for better understanding the genetic implication in sports predisposition.

Also, some tables with results which confirmed the genetic predisposition to sports are useful. 

Author Response

Dear Reviewer,

Thank you for reviewing the manuscript. We value your commitment and contribution to the paper, and your comments have undoubtedly improved its quality. Please find below your comments and our reply. All changes are highlighted in the manuscript file.

Best regards,

The authors

Comments and Suggestions for Authors

Review for the manuscript “Antioxidant Gene Polymorphisms Potentially Predisposing to 2 Exercise ”

In this article, the authors said that there are genetic predisposition to sports, such as the genes that promote oxidative processes, or genes associated with antioxidant processes. 

Comments

Presentation of the article is a litle beat hard to follow. Maybe more figures are useful for better understanding the genetic implication in sports predisposition.

Also, some tables with results which confirmed the genetic predisposition to sports are useful.

Authors’ Response

We decided to divide each chapter into subchapters using subheadings and add a table showing alleles that have been shown to have a proven beneficial effect on physical performance (Introduction, Table 1).

As you can see, we also made minor language corrections.

We have also updated chapter 4 with another study on SOD2 gene polymorphism and exercise implications

The other changes have been dictated by the comments of other reviewers.

Reviewer 4 Report

Comments and Suggestions for Authors

The review is interesting and generally well-written.
According to my opinion, some major concerns should be addressed to improve the impact of the study.
The title is inappropriate because the issue of polymorphisms in genes coding for antioxidant factors is only one of the aspects illustrated mostly in paragraph 4, which is entitled ‘genes responsible...’ and not for example: ‘genetic variants in genes responsible for...’. A more general title could be better.
Moreover, what is missing, although much to be argued, is a discussion of the translatability of the topic addressed in clinical practice. Many genes and proteins presented in the manuscript are crucial in generally regulating the response to oxidative stress and inflammation and redox homeostasis.
This has an important bearing on health status even in the treatment of chronic diseases. Some factors that may be very important in this relate, for example, to fatty acid respiration. There are also data on polymorphisms in this area that have not been taken into account. Furthermore, there are interesting data in this regard that link the modulation of these factors to exercise and the possibility of using it as a therapy that would be interesting to report to improve the introduction and conclusions (e.g. Biomarkers and genetic polymorphisms associated with maximal fat oxidation during physical exercise: implications for metabolic health and sports performance. Eur J Appl Physiol. 2022;122(8):1773-1795. doi:10.1007/s00421-022-04936-0....Cardiac Rehabilitation Increases SIRT1 Activity and β-Hydroxybutyrate Levels and Decreases Oxidative Stress in Patients with HF with Preserved Ejection Fraction. Oxid Med Cell Longev. 2019;2019:7049237).

Comments on the Quality of English Language

English is good:  Minor editing of English language required.

Author Response

Dear Reviewer,

Thank you for reviewing the manuscript. We value your commitment and contribution to the paper. Please find below your comments and our reply. All changes are highlighted in the manuscript file.

Best regards,

The authors

Comments and Suggestions for Authors

The review is interesting and generally well-written.
According to my opinion, some major concerns should be addressed to improve the impact of the study.
The title is inappropriate because the issue of polymorphisms in genes coding for antioxidant factors is only one of the aspects illustrated mostly in paragraph 4, which is entitled ‘genes responsible...’ and not for example: ‘genetic variants in genes responsible for...’. A more general title could be better.
Moreover, what is missing, although much to be argued, is a discussion of the translatability of the topic addressed in clinical practice. Many genes and proteins presented in the manuscript are crucial in generally regulating the response to oxidative stress and inflammation and redox homeostasis.
This has an important bearing on health status even in the treatment of chronic diseases. Some factors that may be very important in this relate, for example, to fatty acid respiration. There are also data on polymorphisms in this area that have not been taken into account. Furthermore, there are interesting data in this regard that link the modulation of these factors to exercise and the possibility of using it as a therapy that would be interesting to report to improve the introduction and conclusions (e.g. Biomarkers and genetic polymorphisms associated with maximal fat oxidation during physical exercise: implications for metabolic health and sports performance. Eur J Appl Physiol. 2022;122(8):1773-1795. doi:10.1007/s00421-022-04936-0....Cardiac Rehabilitation Increases SIRT1 Activity and β-Hydroxybutyrate Levels and Decreases Oxidative Stress in Patients with HF with Preserved Ejection Fraction. Oxid Med Cell Longev. 2019;2019:7049237).

Authors’ Response

 The paper is concerned with genes responsible for shaping the redox response following exercise, and we primarily focused on antioxidant genes because they are the least understood in this context. Hence the chapters on the effects of exercise on the human organism, particularly on the redox balance. The effects of exercise on human health and the application of exercise in medicine are only side topics, introducing with the main theme of the study.

Nevertheless, after thinking about it, we decided that, according to the suggestions, it would not hurt to generalize the title and expand the introduction and conclusions a bit to include such information. We have written it very generally, yet explicitly, while regarding the details, we refer the reader to the literature provided. So we have generalized the title of the study, edited the abstract, added eight sentences in the first paragraph of the introduction, as well as, created a new first paragraph of conclusions (four sentences).

Moreover, in order to improve the readability of the paper, we have divided the chapters into subchapters, added a table in the introduction (Table 1), and the introduction have been slightly edited.

As you can see, we have also made minor language revisions.

We have also updated chapter 4 with another study on SOD2 gene polymorphism and exercise implications

Other changes have been dictated by the comments of other reviewers.

Round 2

Reviewer 4 Report

Comments and Suggestions for Authors

The authors responded correctly to the reviewers' requests.

Comments on the Quality of English Language

English Language is ok.   Minor editing of English is required.